# Recent Warming Trends in the Arabian Sea: Causative Factors and Physical Mechanisms

**Jiya Albert** [1] , **Venkata Sai Gulakaram** [2] , **Naresh Krishna Vissa** [2] , **Prasad K. Bhaskaran** [1,*] and **Mihir K. Dash** [3]

1    Department of Ocean Engineering & Naval Architecture, Indian Institute of Technology Kharagpur, Kharagpur 721 302, India
2    Department of Earth and Atmospheric Sciences, National Institute of Technology, Sundargarh District, Rourkela 769 008, India
3    Centre for Oceans, Rivers, Atmosphere and Land Sciences, Indian Institute of Technology Kharagpur, Kharagpur 721 302, India
*    Correspondence: pkbhaskaran@naval.iitkgp.ac.in; Tel.: +91-3222-283772

**Abstract:** In recent years, and particularly from 2000 onwards, the North Indian Ocean (NIO) has been acting as a major sink of ocean heat that is clearly visible in the sub-surface warming trend. Interestingly, a part of the NIO—the Arabian Sea (AS) sector—witnessed dramatic variations in recent sub-surface warming that has direct repercussion on intense Tropical Cyclone (TC) activity. This study investigated the possible causative factors and physical mechanisms towards the multi-decadal warming trends in surface and sub-surface waters over the AS region. Responsible factors towards warming are examined using altimetric observations and reanalysis products. This study used ORAS5 OHC (Ocean Heat Content), derived meridional and zonal heat transport, currents, temperature, salinity, Outgoing Longwave Radiation (OLR), and air-sea fluxes to quantify the OHC build-up and its variability at water depths of 700 m ($D_{700}$) and 300 m ($D_{300}$) during the past four decades. The highest variability in deeper and upper OHC is noticed for the western and southern regions of the Indian Ocean. The warming trend is significantly higher in the deeper regions of AS compared to the upper waters, and relatively higher compared to the Bay of Bengal (BoB). Increased OHC in AS show good correlation with decreased OLR in the past 20 years. An analysis of altimetric observations revealed strengthening of downwelling Kelvin wave propagation leading to warming in eastern AS, mainly attributed due to intrusion of low saline water from BoB leading to stratification. Rossby wave associated with deepening of thermocline warmed the southern AS during its propagation. Heat budget analysis reveals that surface heat fluxes play a dominant role in warming AS during the pre-monsoon season. Increasing (decreasing) trend of surface heat fluxes (vertical entrainment) during 2000–2018 played a significant role in warming the southeastern sector of AS.

**Keywords:** ocean heat content; heat transport; Kelvin wave; Rossby wave; Indian Ocean Dipole

## 1. Introduction

The Arabian Sea (AS), a semi-marginal sea located in the western North Indian Ocean (NIO) basin had recently experienced enhanced Tropical Cyclone (TC) activity compared to the eastern sector, the Bay of Bengal (BoB) [1–4]. Most of the occurrences are focused over the onset phase of the Indian Summer Monsoon [2], and the co-existence of positive IOD and La Niña events has contributed to more intense cyclones in the recent decades [1]. Enhanced convective activity over this basin, along with the formation and intensification of intense TCs can have profound impact on the socio-economic balance of coastal community in the Indian sub-continent [5]. Tropical cyclones are disastrous extreme weather event phenomena that can result in widespread destruction to life and property during its landfall. It can lead extreme water level and wind-wave conditions [6–11] and also change the upper ocean characteristics [12–15] having implications on coastal vulnerability [16–19]. A recent

study [20] indicated that intensity of severe cyclonic storms in the NIO region showed an increasing trend in the past four decades. Based on historical archived best-track cyclone records, the scientific community is well aware that cyclone frequency is 5:1for the BoB and AS regions located in the NIO region, respectively. However, in a changing climate and in particular during the recent decades, the AS basin is no longer considered as a less cyclone-active in terms of TC activity. The pre- (March-May) and post-monsoon (October– December) seasons witnessed intense TC activity over the AS region. Some of the examples of the cyclonic storms witnessed over AS region are tabulated in Table 1.

**Table 1.** Summary of cyclonic storms witnessed over Arabian Sea region.

| Intensity | Name of the Cyclone | Time Period |
|---|---|---|
| Super Cyclonic Storms (SuCS) | GONU | 1–7 June 2007 |
| | KYARR | 24 October–3 November 2019 |
| Extremely Severe Cyclonic Storms (ESCS) | ARB 01 | 21–28 May 2001 |
| | NILOFER | 23–31 October 2014 |
| | CHAPALA | 28 October–4 November 2015 |
| | MEGH | 4–10 November 2015 |
| | TAUKTE | 14–19 May 2021 |
| Very Severe Cyclonic Storm (VSCS) | PHET | 31 May–7 June, 2010 |
| Severe Cyclonic Storm (SCS) | NISARGA | 1–4 June 2020 |

Strong energy interactions through air-sea enthalpy (latent and sensible heat) fluxes are the critical mechanisms driving the generation and propagation of TCs [21–23]. Observed warming over NIO and specifically the region confined over the less cyclone-active AS basin with enhanced convective activity is attracting the attention of researchers [24,25]. Prior studies have focused exclusively on the western tropical Indian Ocean of AS signifying warming at a faster rate as compared to rest of the tropical Indian Ocean [20,26,27]. A study by [28] demonstrated that cyclogenesis over AS during the post-monsoon season is modulated by both the monsoon rainfall and El Niňo with a support of positive Indian Ocean Dipole (IOD) events. It is noteworthy that a single phenomenon hardly explains the observed variability over the AS region. Recently, Ref. [29] demonstrated that apart from several known atmospheric forcing effects, the inter-annual variability of OHC (Ocean Heat Content) could influence the post-monsoon AS cyclogenesis. Additionally, the salinity variations existing over the domain partially modulates the OHC of this region. Their study provided valuable information on the heat exchange mechanism between the southwest Indian Ocean (SWIO) and AS, that induces variability in the AS OHC. In addition, their analysis [29] also attributed the recent trend for increased OHC (from 2011 onwards) with elevated sea-surface carbon levels over the AS region. Moreover, it is remotely influenced by the heat transport mechanism facilitated largely by the variability in equatorial current flow patterns [30]. It is interesting to note that the projected Indian Ocean (IO) warming trend is faster than the input heat sources attributed from direct air-sea heat flux. Significant weakening of the Agulhas current system and its transport in a warming climate could further reduce the outward heat transport from the southern exit of the Indian Ocean [30]. Recently, the studies also confirmed two to three times increased biological productivity in AS waters with the subsequent occurrence of TCs [31]. On a longer time-scale, the emission of $CO_2$ gas from the ocean to the atmosphere could increase from 2 to 6 times. Findings also demonstrate the possibility of more frequent TCs in the future that could lead to oxygen deficiency in the AS. The release of more $CO_2$ into the atmosphere in a longer-term perspective increases the possibility for more warming [31]. Conducive ocean-atmosphere conditions in the AS basin linked with recent warming trends has a direct bearing on the TC activity. Based on the above discussions, and keeping in view the importance of this

problem, it is pertinent to explore the different pathways of heat transport influencing the surface and sub-surface waters of AS that still remain as an open question warranting a detailed study to understand the future TC activity over this basin. This study aims to understand the causative factors and responsible physical mechanisms of the oceanic processes that contribute to enhanced heat content in the AS region.

### 1.1. Arabian Sea Warming and Enhanced Cyclone Activity

The Indian Ocean (IO) has experienced rapid warming since 1998, which is intrinsically associated with an increased water mass exchange by the Indonesian Throughflow (ITF) [32–34]. In addition, prior studies have established that any variations in a warming environment can lead to an increase in the frequency and intensity of TCs [35–37]. For example, the study by [38] based on ensemble simulations from 15 coupled GCMs (General Circulation Models) showed a 4.6% increase in the TC potential intensity over the AS region. Analyzing the TC data from 1970 to 2007, Ref. [24] reported a negative-fold increase in the occurrence of TCs after 1995, the period when AS was undergoing a regional climate shift. Subsequently, Ref. [39] proposed that the recent increase in the intensity of pre-monsoon (May–June) TCs over AS is a consequence due to reduction in the basin-wide vertical wind shear attributed due to simultaneous increase in the anthropogenic black carbon and sulfate emissions. In addition, there are other studies that strongly signify frequent occurrence of intense cyclones over the AS region that can pose significant risk and vulnerability to the coastal regions. A recent study by [20] postulated an increase trend (~ 16.75%) in the intensity of SCS (Severe Cyclonic Storms) that formed over NIO during the past four decades. The Power Dissipation Index (PDI) also showed a correlation of 0.58 with SCS during the period 1979–2019 [3]. The above-mentioned studies clearly decipher that warming in the AS is factual and that has a significant bearing on enhanced severity of TCs.

### 1.2. Tropical Cyclone Formation and Equatorial Wave Activity

The important contributions from TC-induced ocean mixing towards the variation in ocean heat transport and circulation remained as an indignation [40–42]. Mixing is driven by the strong surface wind stress induced by the TC vortex motion. Further, there were disagreements, namely the TC induced circulation mixing pumps warm water from surface to the deeper layers. The equator-poleward (meridional) warm water transport contributes towards the variation in Ocean Heat Transport (OHT).

The important contributions outlined above are reinforced by the remote forcing effects on the mid-latitude weather systems attributed due to inter-basin scale SST (Sea Surface Temperature) and its variability [35,43]. The major fuel triggering the genesis and propagation characteristics of TCs is the heat transfer from the ocean [44]. Generation of this fueling energy is initiated from the surface heat fluxes and moisture from the ocean. The basic state of the atmospheric conditions varies due to this energy transfer enhancing its potential for convection and precipitation [45]. On the contrary, a negative feedback mechanism is initiated parallelly by the cyclone-induced wind stress due to turbulent mixing of the upper ocean layers, that eventually cools the ocean surface by Ekman pumping [46–49]. These studies provide an inevitable link between the thermodynamic properties of the ocean with TCs throughout its entire life span. There are several studies that postulated a strong relationship existing between TC formation and the equatorial wave activity [50–56]. The major contributions were provided by (i) Mixed Rossby Gravity Waves, (ii) Equatorial Rossby Waves, (iii) Kelvin Waves, and (iv) Madden Julien Oscillation [57]. These waves appear to enhance the local circulation patterns by increasing the organized upward vertical motion, low-level vorticity at the genesis location, and modulate the vertical shear [51]. In this study, the inevitable factors that contribute to AS warming and the possible physical mechanisms are explored. Using this conceptual evaluation, the present study aims to provide a plausible explanation on the recent intensification of TCs over the AS region. The first objective explores the sensitivity of ocean warming

phenomenon over the AS and its comparison with neighboring regions in the NIO. Further, a comprehensive study is reported to establish the impact of equatorial waves on TC activity over the AS region.

### 1.3. Role of Kelvin Wave Activity

The NIO is a unique ocean basin that experience seasonal reversal of the monsoonal wind system having a direct implication on the ocean circulation aspect, which makes the NIO quite distinctive compared to the other ocean basins [58,59]. In the Equatorial Indian Ocean (EIO), the zonal winds are semi-annual and seasonal in nature triggering the fluctuations in sea surface heights over both the central and eastern equatorial regions [60–62]. The easterly (westerly) winds over this region are favorable towards the formation of equatorially trapped upwelling (downwelling) Kelvin waves that propagate towards the eastern equatorial Indian Ocean. After reaching the eastern boundary, these waves propagate as coastally trapped waves along the BoB coast, also known as the coastal Kelvin waves [60–63]. During the course of propagation, the westward propagating Rossby waves are radiated by the Kelvin waves influencing the circulation over the NIO region [61–65]. In the EIO, there are two pair of alternate upwelling and downwelling Kelvin waves that modulate the circulation aspects of BoB and AS [58,60,61]. Among these Kelvin waves, the second downwelling Kelvin waves that propagate during October–December in BoB, are the only waves that travel to AS influencing the west Indian coastal current [58,60,61,63,64]. A sea-level high is formed in the Southeastern Arabian Sea (SEAS) due to the propagation of Kelvin and radiated Rossby waves, which is also known as Lakshadweep high [58,66,67]. Ref. [68] postulated that the intrusion of low salinity waters during November–December forms the near-surface stratification, which sets the stage for formation of warm pool in the following months. With the evolution of time, this will extent spatially and spread southwestward towards the Somalia coast [60,68,69].

To address the physical mechanism behind the projected warming in the Arabian Sea (AS), the current study has considered various physical mechanisms responsible for warming trend in the Arabian Sea region using altimetric observations and reanalysis products. Section 2 contains the relevant information regarding the data sets and methodology used in the study. This follows the salient observations, results and discussion covered in Section 3. Finally, the conclusions and important outcomes of the study are presented in Section 4.

## 2. Data and Methods

### 2.1. Data

This study uses the datasets pertaining to ocean heat, derived heat transport, currents, temperature, salinity, OLR and air-sea fluxes that can provide valuable insights into the underlying dynamics that cause the variability in convective activity over the AS basin. The ocean currents, temperature, and salinity datasets were obtained from the Ocean Reanalysis System 5 (ORAS5, monthly). ORAS5 is an eddy-permitting ocean and sea-ice ensemble reanalysis system that provides the global ocean and sea-ice conditions from 1979 to the present [70]. Daily Outgoing Longwave Radiation (OLR) data is obtained from the National Oceanic and Atmospheric Administration (NOAA) [71] with a spatial resolution of 2.5°, available at (http://apdrc.soest.hawaii.edu, accessed on 6 September 2021). Daily air-sea fluxes such as the net heat flux, latent heat flux, sensible heat flux, and wind stress data were obtained from the TropFlux project of INCOIS (Indian National Centre for Ocean Information Services) [72,73]. Evaporation data is obtained from the OA flux project of the Woods Hole Oceanographic Institution (WHOI) having a spatial resolution of 1° × 1° for the period 1985–2018 [74].

This study investigated the intra-seasonal heat transport variability using an objective index that identifies the ocean Equatorial (Kelvin and Rossby) waves (EW) using the satellite observations of Sea Surface Height (SSH) [75–78]. The daily SSH anomalies data having a spatial resolution of 0.25° and covering a period of 26 years (1993–2018) were

used in this study. SSH anomaly data were obtained from the Copernicus Marine and Environment Monitoring Service (CMEMS, https://marine.copernicus.eu/, accessed on 8 September 2021) to demonstrate the decadal variations in the propagation characteristics of Kelvin and Rossby waves. The current methodology is superior and versatile because it exhibits the systematic relationships between intra-seasonal modes in the atmosphere and ocean. The procedure involved in evaluating Equatorial Waves (EW) and the derived teleconnections hold regardless of the identifying index used.

## 2.2. Methodology

The initial analysis focused on the assessment of relative warming in AS and its comparison with other sub-domains over the NIO region. The study domain is sub-divided into seven sub-regions [79] based on the geographical boundaries (as listed in Table 2 and Figure 1). To address the physical mechanism attributed to projected warming in the Arabian Sea (AS), the current study has considered multiple aspects observed by different researchers. Skillful predictions of OHC in the upper 300 m and 700 m across dynamical environment contributes to accurate surface predictions such as Sea Surface Temperature (SST) [25]. Higher warming in the AS compared to other ocean basins have been reported in previous observations [80], in which the largest projected and observed trends in relative SST and potential intensity were found over the tropical part of this region. Another study has also proven the influence of Atlantic Ocean SST on the variations in AS SST [81]. Large scale circulation events like La Niña are usually associated with the accumulation of heat in the western tropical Pacific which is further transported to the Indian Ocean sector such as AS and BoB [82]. The occurrence of much deeper thermocline depth leads to the vertical redistribution of heat in the Indo-Pacific Ocean basin [83]. Similarly, the heat budget over the Pacific and Indian Ocean basins are affected by the transport of heat through the Indonesian through flow [84–86]. Therefore, the effective warming over the AS basin needs a detailed analysis over both AS and BoB regions (depicted as Regions 1 and 2), the Pacific Ocean Component-South China Sea (Region 3), tropical regions A and B (Regions 4 and 5), adjacent open ocean west and east sector from the South Indian Ocean (Regions 6 and 7).

**Table 2.** Details of sub-domains in the Indian Ocean region used for the study.

| Label | Name | Region | Domain |
|---|---|---|---|
| ARB | Reg 1 | Arabian Sea | EQ: 30° N; 30° E: 78° E |
| BoB | Reg 2 | Bay of Bengal | EQ: 30° N; 78° E: 105° E |
| SCS | Reg 3 | South China Sea | EQ: 30° N; 105° E: 120° E |
| SIO (A) | Reg 4 | Tropical South Indian Ocean—A | 35° S: EQ; 30° E: 78° E |
| SIO (B) | Reg 5 | Tropical South Indian Ocean—B | 35° S: EQ; 78° E: 120° E |
| WSO | Reg 6 | Western extra-tropical south Indian Ocean and the Southern Ocean (south of 60° S) | 63° S: 35° S; 30° E: 76° E |
| ESO | Reg 7 | Eastern extra-tropical south Indian Ocean and the Southern Ocean (south of 60° S) | 63° S: 35° S; 76° E: 120° E |
| SEAS (A) | D1 | South eastern Arabian Sea—A | 2° N: 7°N; 73° E: 80° E |
| SEAS (B) | D2 | South eastern Arabian Sea—B | 4° N: 14°N; 68° E: 78° E |

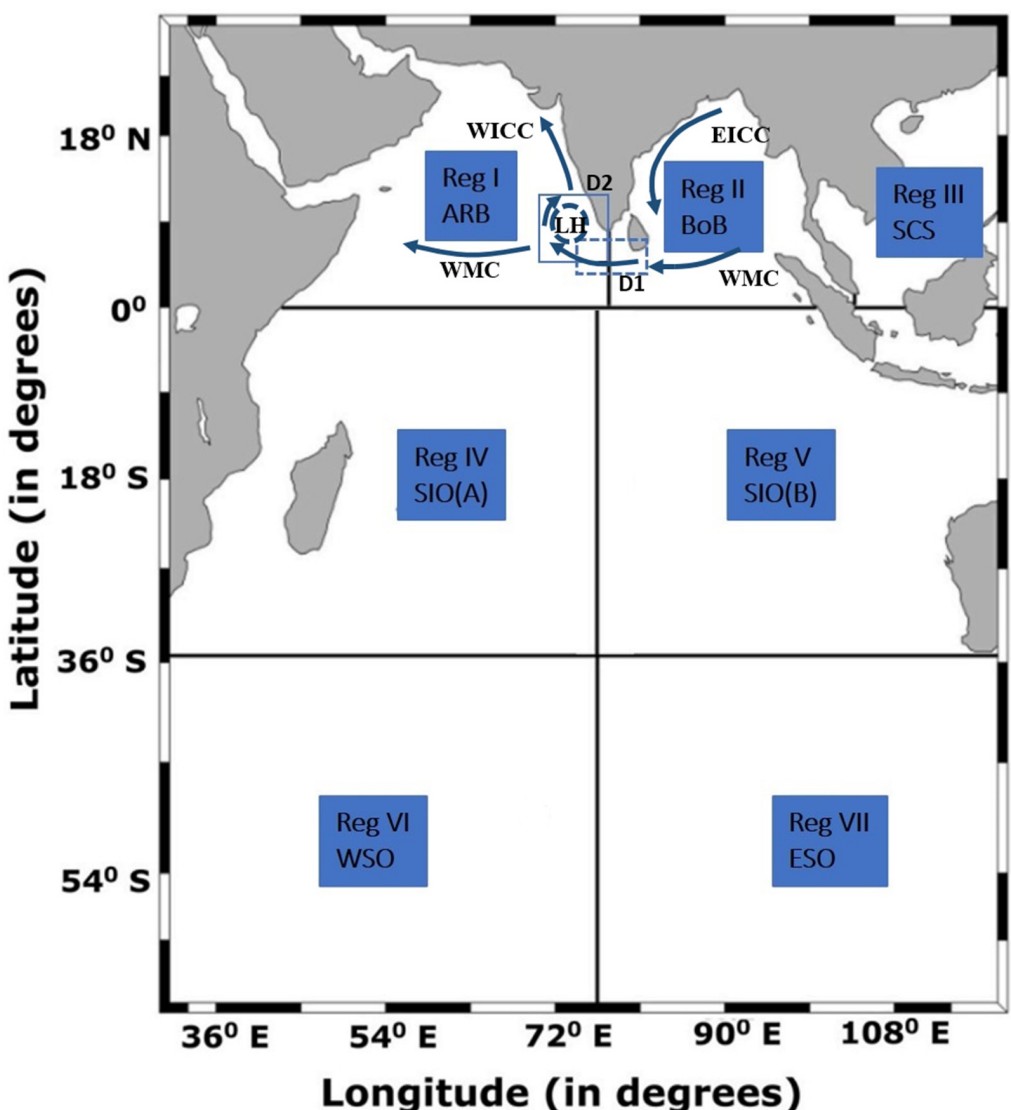

**Figure 1.** The map of Northern Indian Ocean region showing the study regions marked I-VII and D1 (2° N–7° N; 73° E–82° E) and D2 (4° N–14° N; 68° E–78° E) along with the schematic representation of circulation in the NIO during winter monsoon. WICC—West Indian Coastal Current, EICC—East Indian Coastal Current, WMC—Winter Monsoon Current, and LH—Lakshadweep High.

Sub-surface OHC supports the TC formation and its intensification. Maximum energy supplied by the oceans towards the convective activity in the atmosphere is available within the upper 300 m water depth. Heat content in the relatively deep oceanic layer (700 m) represents the distribution and inter-basin scale heat exchange between the adjacent water bodies. Therefore, two independent datasets: (i) for OHC integrated over the upper 300 m ($D_{300}$) and (ii) upper 700 m of the oceanic layer ($D_{700}$) were used in this study to identify the amount of ocean warming. The $D_{300}$ is used to assess the amount of sub-surface warming over the different sub-domains and $D_{700}$ is used to determine the warming over deeper oceanic layers during the period 1979–2020.

The OHC was estimated using the mathematical formulation:

$$H = \rho C_p \int_{h1}^{h2} T(z)dz \tag{1}$$

where $T(z)$ represents the temperature as a function of water depth ($z$) integrated over the column from surface to 300 m or 700 m water depths. Thereafter, the OHC transport is

estimated in terms of meridional and zonal components for the study period. Meridional and zonal heat transport across the latitude from BoB to AS are estimated by using the following equations described by [87–89].

$$\text{Meridional Transport} = c_p\, \rho \int_{x1}^{x2} \int_{z}^{0} \theta v\; dz\; dx \tag{2}$$

$$\text{Zonal Transport} = c_p\, \rho \int_{y1}^{y2} \int_{z}^{0} \theta u\; dz\; dy \tag{3}$$

where the terms $x$, $y$ represent the zonal and meridional coordinates, respectively, $z$ is the vertical coordinate, $u$ and $v$ are the zonal and meridional velocities, respectively. The terms $\theta$, $\rho$, $c_p$ represents the potential temperature, seawater density, and the specific heat capacity of seawater, respectively.

The ocean mixed-layer heat budget analysis is estimated to understand the processes responsible for increasing the warming trend in the sub-surface water depths. The heat budget is computed by using the methodology described by [90–92].

$$\frac{\partial T}{\partial t} = \frac{Q_{net}}{C_p \rho h} - \left[ u \frac{\partial T}{\partial x} + v \frac{\partial T}{\partial y} \right] - \frac{w_e \Delta T}{h} \tag{4}$$

In the presence of Barrier Layer Thickness (BLT):

$$w_e = \frac{\Lambda_w \left( c_1 u_*^3 - c_2 Bh \right)}{-\beta g h \Delta S} \tag{5}$$

In the absence of Barrier Layer Thickness (BLT):

$$w_e = \frac{\Lambda_w \left( c_1 u_*^3 - c_2 Bh \right)}{\alpha g h \Delta T} \tag{6}$$

where,

$$B = \frac{\alpha \times qq}{\rho C_p} + \beta S_0 (P - E) \tag{7}$$

The four terms in the above Equation (4) represents the temperature tendency, surface heat flux (atmosphere forcing), horizontal advection, and the entrainment flux. The variable $T$ represents the mixed layer averaged temperature, $Q_{net}$ is the near-surface net heat flux, $C_p$ is the specific heat constant, $\rho$ is the density of seawater, $h$ is the mixed layer depth, $u$ and $v$ are the zonal and meridional velocities, respectively. The term $\Delta T$ represents the difference between the mixed layer averaged temperature and the temperature at the base of the mixed layer. The entrainment velocity is represented by $w_e$ which is parameterized based on the presence and absence of barrier layer [15–91].

The estimations of uncertainty in the linear trends were carried out using the Confidence Interval (CI). This is a statistical technique that determines the probability of a calculated interval containing a true value. Trend statistics were estimated with 95% CI, and it provides information on how reliable the estimation is connected, with respect to the other samples collected. Using the bootstrapping method, samples are replaced with data points existing in a common dataset that leads to more accurate and efficient results related to CI. The method includes a t-test to determine the statistical significance, and bootstrapping method can provide the confidence interval value for the estimated slope of a linear trend. The interval indicates the range (upper bound and lower bound) of credible values for the slope, given the data and assumptions used in the analysis. For the desired parameters of OHC and OLR, the CI is estimated to have narrow confidence intervals. The results are more desirable since this provides us with a narrow range of values that contain the linear trend. All the data sets are tested using the t-test and found to be statistically significant.

## 3. Results and Discussion

### 3.1. Analysis of Ocean Warming at Various Sub-Domains in the NIO

This study performed a detailed analysis to understand the variability in OHC utilizing the ORAS5 data for both the upper 300 m ($D_{300}$) and 700 m ($D_{700}$) water depths during the period 1979–2020. The spatial averaged OHC values were considered in the study domain that covers seven different sub-domains. Analysis was carried out on the time-series distribution of OHC data at $D_{300}$ (Figure 2) and $D_{700}$ (Figure 3) at all the seven sub-domains that represents the northern, central, and southern Indian Ocean regions.

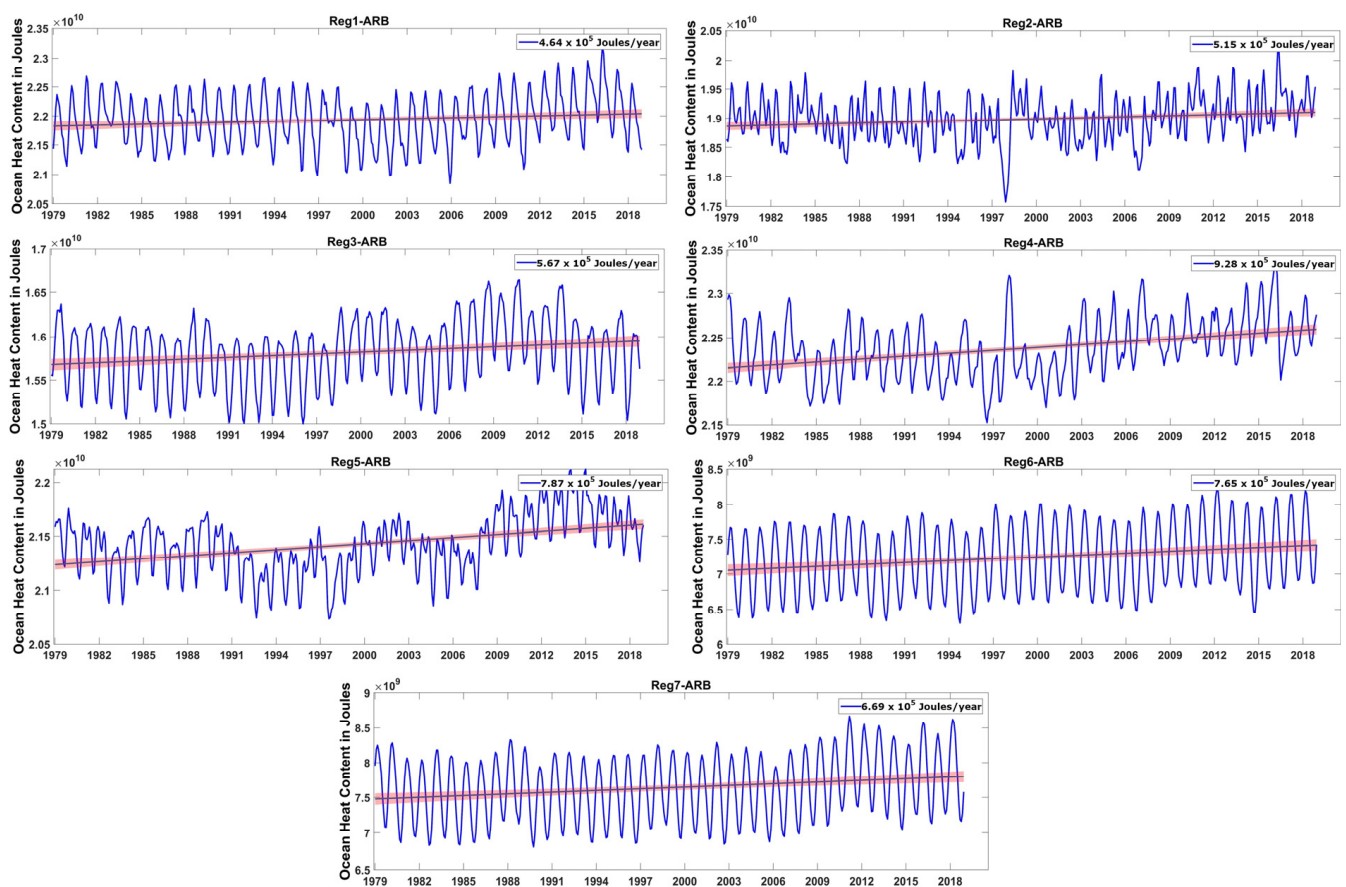

**Figure 2.** Variability in ORAS5 OHC at 300 m along with the corresponding confidence interval (red patches) at different sub-domains of the Indian Ocean region (1979–2020).

Results suggest that the trends in general exhibited by $D_{700}$ are relatively higher compared to $D_{300}$ at all the seven locations (Figures 2 and 3). Maximum trend value of $D_{300}$ is observed at Reg 4 (Figure 2) that corresponds to the tropical South Indian Ocean connecting the AS portion and minimum for the ARB region (Figure 2). However, the highest variability in $D_{700}$ is noticed at Reg 6 (Figure 3f) that corresponds to the western extra-tropical South Indian Ocean and the Southern Ocean regions (south of 60° S). The oceanic layers exhibit a uniform warming throughout the depth ($D_{700}$ and $D_{300}$) mostly over the western side of the Indian Ocean and the Southern Ocean. Additionally, it supports findings from the prior studies that reported on significant weakening of the Agulhas current transport, which reduces the southward heat transport out of the Indian Ocean resulting in heat accumulation over the southern regions [30].

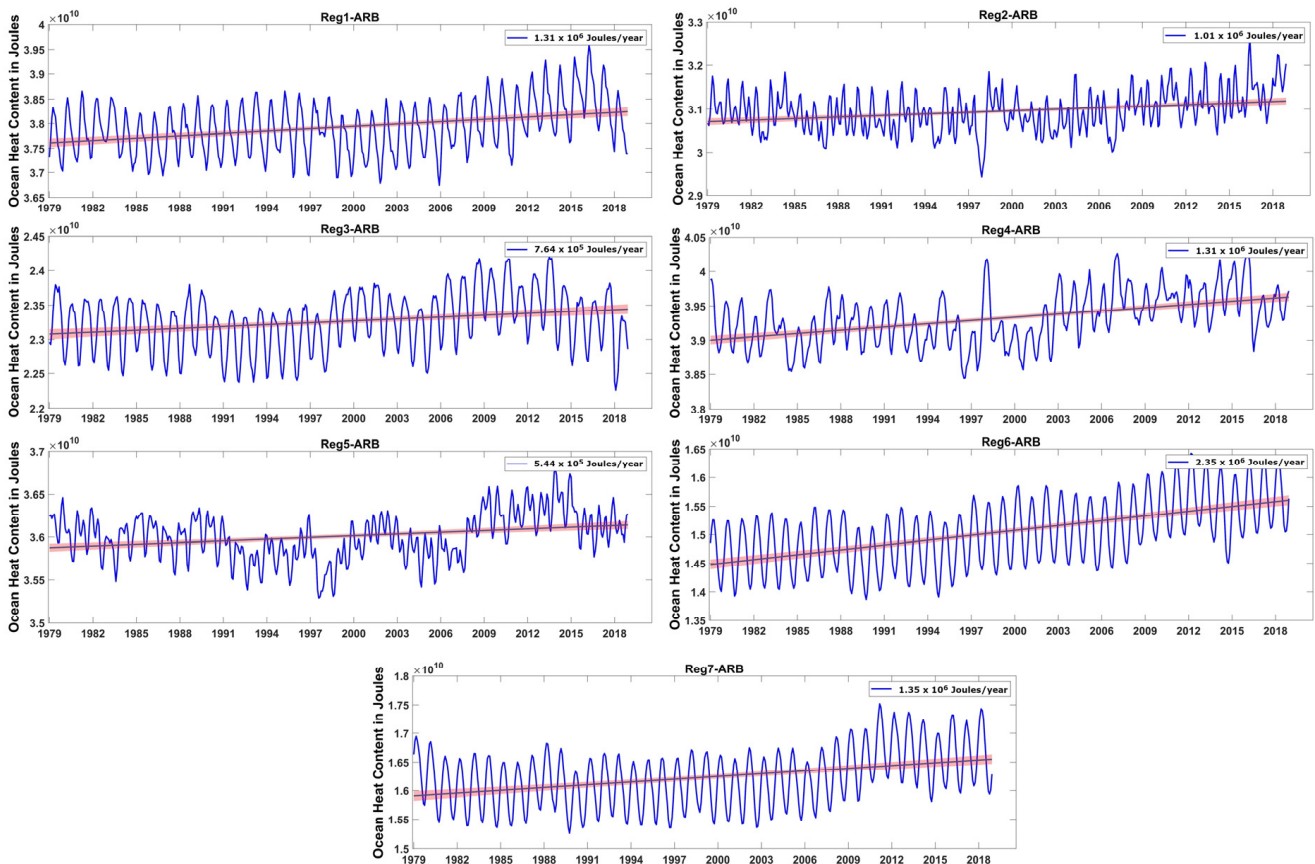

**Figure 3.** Variability in ORAS5 OHC at 700 m along with the corresponding confidence interval (red patches) at different sub-domains of the Indian Ocean region (1979–2020).

To determine the requisite conditions that prevail over both the AS and BoB basins, it is required and necessary to perform a comparison of the OHC time series. The time-series trend in OHC considering two epochs (before and after the year 2000) over both the AS and BoB basins at $D_{300}$ and $D_{700}$ showed a distinct 10-year cycle (Figure 4). The top panel (Figure 4a) represents the $D_{300}$ OHC distribution, and the sub-plots are for Reg 1 in AS (top-panel) and BoB (bottom-panel). It is clearly seen from Figure 4a, that prior to the year 2000, the OHC trend in the AS and BoB were decreasing at a rate of $-0.45 \times 10^6$ J yr$^{-1}$ and $-0.81 \times 10^6$ J yr$^{-1}$, whereas for the post-2000 period in both these basins, the OHC trend showed an increase at a rate of $2.8 \times 10^6$ J yr$^{-1}$ and $2.3 \times 10^6$ J yr$^{-1}$ with a higher increasing rate in the AS sector compared to BoB. Similarly, for the $D_{700}$ OHC distribution (Figure 4b), it is evident that prior to the year 2000, the OHC trend in AS and BoB were decreasing at a rate of $-0.072 \times 10^6$ J yr$^{-1}$ and $-0.94 \times 10^6$ J yr$^{-1}$, whereas for the post-2000 period in both these basins, the OHC trend showed an increase at a rate of $4.32 \times 10^6$ J yr$^{-1}$ and $3.78 \times 10^6$ J yr$^{-1}$ with an higher incremental rate in the AS sector compared to BoB. It is interesting to note that the warming trend in AS since 1979 until the present (2020) has been consistent and the OHC warming trend for $D_{700}$ is much higher ($4.32 \times 10^6$ J yr$^{-1}$) as compared to the BoB ($3.78 \times 10^6$ J yr$^{-1}$). The increased warming for AS in the present decade (2000–2020) is about $0.54 \times 10^6$ J yr$^{-1}$ higher compared to BoB and this enhanced warming in AS has a consequence on the TC activity.

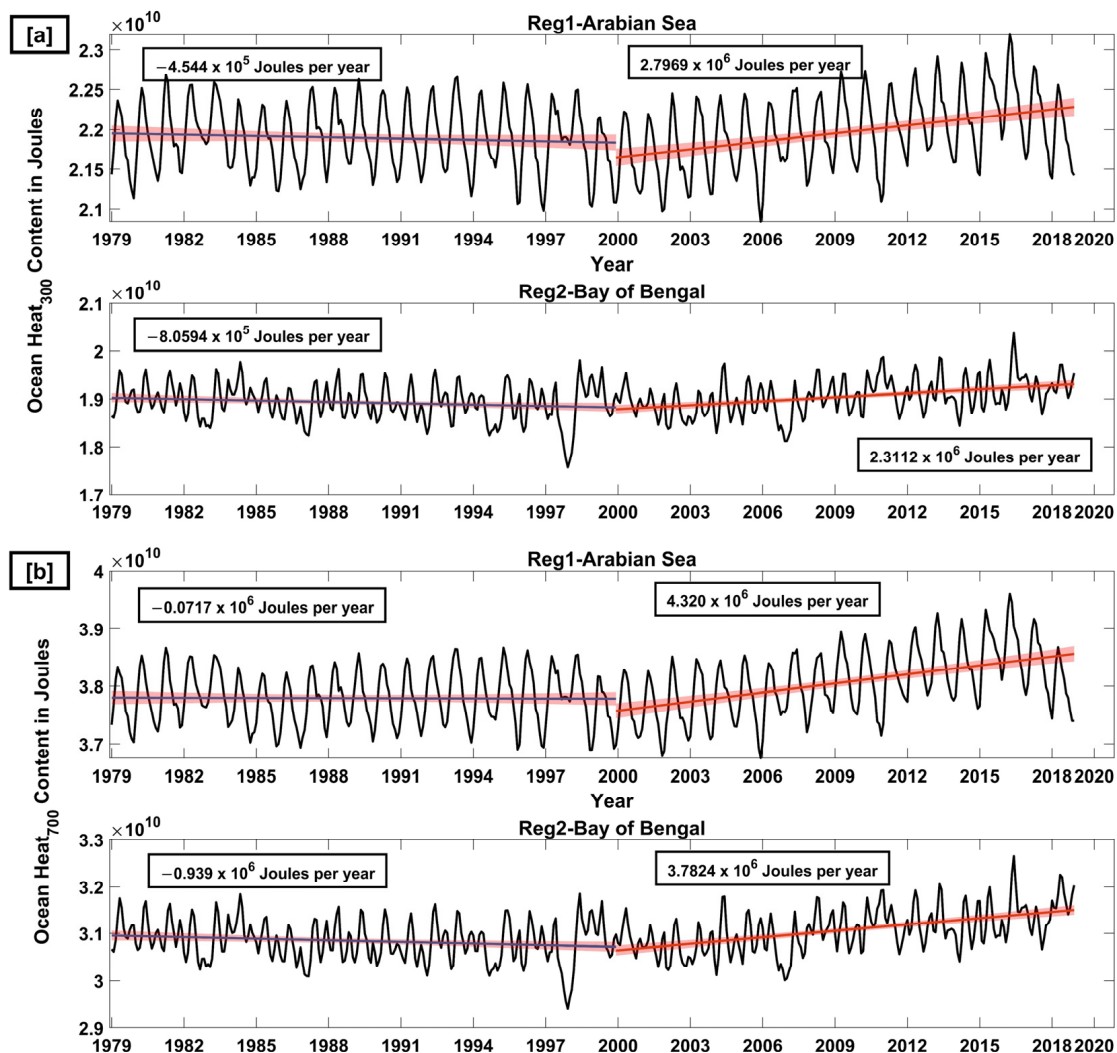

**Figure 4.** Variability in ORAS5 OHC at (**a**) D300 (top panel) and (**b**) D700 (bottom panel) along with the corresponding confidence interval (red patches) over the Arabian Sea and Bay of Bengal region during the period 1979–2020 (trend analysis separated before and after the year 2000).

The trends mentioned above clearly decipher that warming is consistent in the NIO basin, and the heat build-up in both the AS and BoB basins is a matter of concern. It is evident from the analysis period 2000–2020 exhibiting significant warming trends at both $D_{300}$ and $D_{700}$. For deeper water depths $D_{700}$ the warming is about 1.54 (1.64) times higher compared to $D_{300}$ for the AS (BoB) (Figure 4a) clearly indicating heat accumulation in deeper water depths.

Considering the deeper water, the warming trend at $D_{700}$ over AS is almost 12.5% higher compared to the BoB. Similarly, the AS experiences almost 17.37% warming at $D_{300}$ that is about half the OHC at $D_{700}$ (Figure 4a,b). OHC depends not only on the heat inflow-outflow exchanges but also on the longwave radiative fluxes.

The clear indicator for global heating is contributed by the energy imbalance between the incoming solar radiation and outgoing longwave radiation due to the increase in the heat-trapping gases emitted by anthropogenic activities [92–94]. The residual net sea surface heat fluxes ($Q_{net}$) comprises the heat due to global warming in the global ocean basins [95]. The OHC anomaly is a better metric for the impact due to projected climate change. The present study carried out the estimation of trend analysis in $Q_{net}$ (Figure S1), air temperature at 2 m height (Figure S2) and wind speed at 10 m (Figure S3) over the AS and BoB basins. The ocean heat content warming is substantiated by the decreasing net sea

surface heat fluxes and increased air temperature along with the decreased wind speed. Prior studies indicate that the level of thermal energy progression reaches up to the 2000 m depth of ocean since the late 1950s until date. In this study, we examine the sensitivity of Outward Longwave Radiation (OLR) towards the variation of OHC in both AS and BoB basins (Figure 5). Increased heat storage in the AS waters has a correlation with decreased OLR ($-0.109$ W m$^{-2}$) during the past two decades (Figure 5a). Instead, a less significant decreasing trend in OLR values ($-0.0149$ W m$^{-2}$) are seen over the BoB for the same period (Figure 5b).

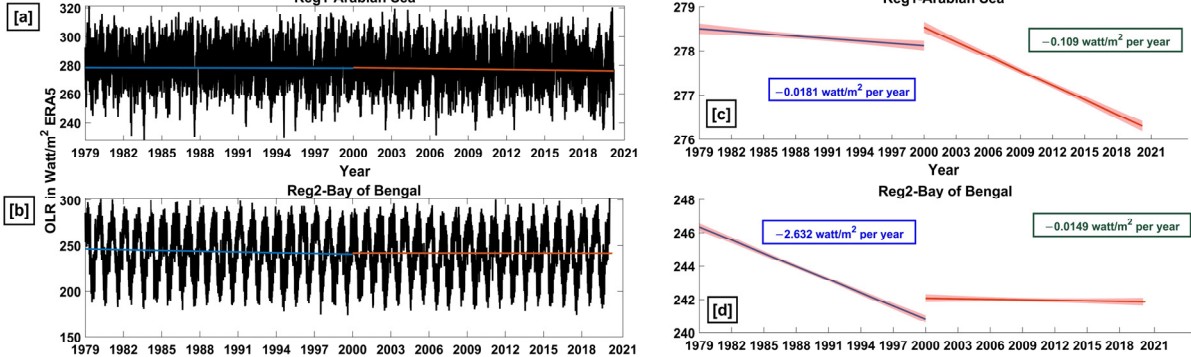

**Figure 5.** Trend of Outgoing Longwave Radiation (W m$^{-2}$) along with the corresponding confidence interval (red patches) over the (**a**) Arabian Sea and (**b**) Bay of Bengal region during 1979–2020. The magnified view of the OLR trends on a two decadal scale are shown in (**c**) for Arabian Sea, and (**d**) for Bay of Bengal.

*3.2. Role of Second Downwelling Kelvin Wave*

The spatial variation in SSHA representing the propagation characteristics of second downwelling Kelvin wave in the NIO region during October–December is shown in Figure 6. The propagation of the second downwelling Kelvin wave (high SSHA) initiates from the EIO in October (Figure 6a) due to the favorable conditions formed by the presence of westerly winds over the EIO [60,62]. The propagation Kelvin wave reaches the southern tip of Sri Lanka by November leading to high sea level (Figure 6b) along the coast of BoB as a coastally trapped Kelvin wave. This high sea level further extends westward during December with an increase in its intensity, reaches the Lakshadweep Sea and forms increased sea level known as Lakshadweep High (Figure 6c) [58,67]. This formation of high further leads to the development of high SST in the SEAS [69,96]. Ref. [97] proposed the importance of the Kelvin wave propagation in generating the favorable conditions for the formation of SST high in the SEAS. High SSHA associated with the warm SST extends further southwestward towards Somalia coast as a function of time [60,69]. Furthermore, from Figure 6, it is evident that strength of downwelling Kelvin wave has significantly increased from first altimetric decade (1993–2000) to the recent decades (2011–2018). The time series plot of longitude versus time for SSH anomaly averaged for the equatorial region ($-2.5°$ S to $2.5°$ N) is shown in Figure 7. It clearly depicts the eastward propagation and decadal increase in the strength of the Kelvin wave in recent years.

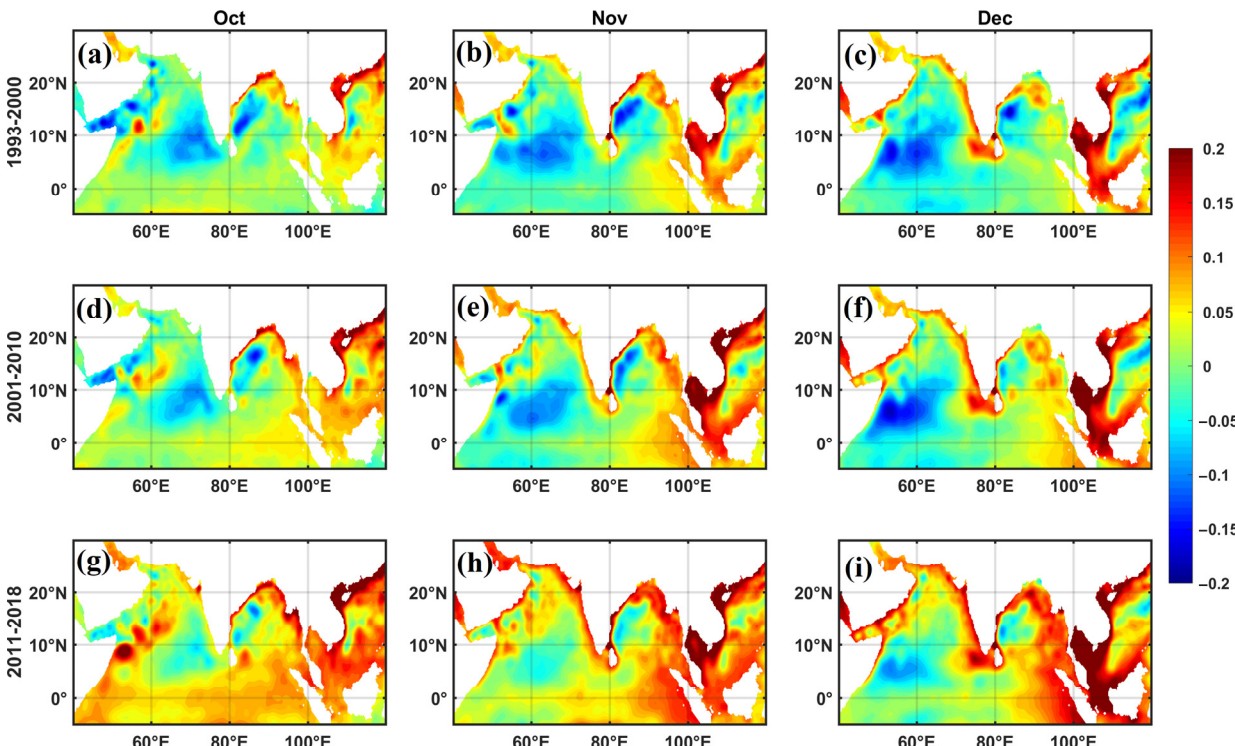

**Figure 6.** The decadal variation in SSHA (m, color shading) representing the propagation of second downwelling Kelvin wave in the North Indian Ocean during October, November and December.

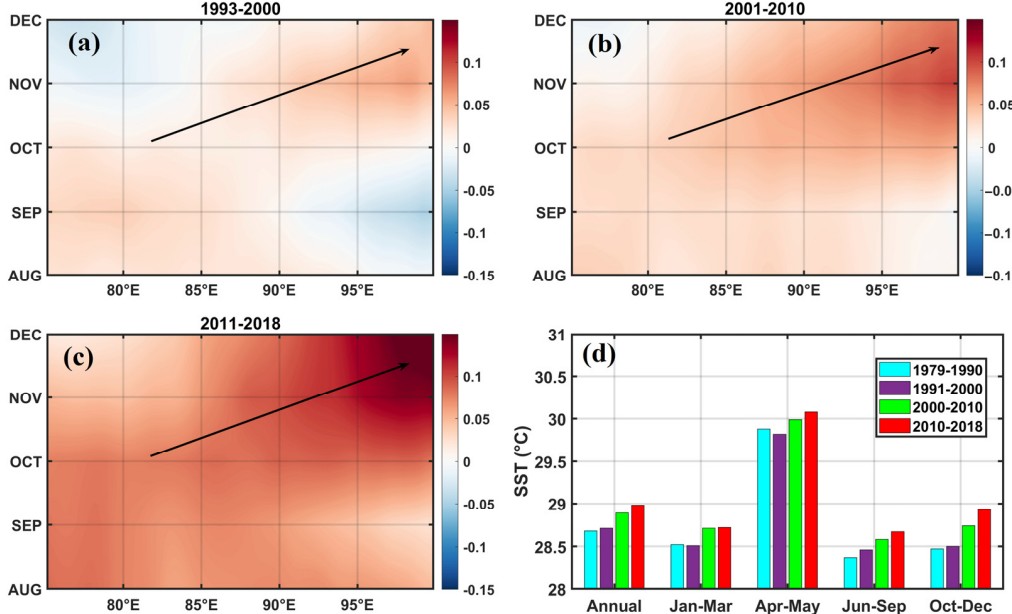

**Figure 7.** (**a–c**) SSHA decadal variation in longitude versus time plot averaged for the latitudinal band (−2.5° S to 2.5° N). The black arrow represents the propagation of the Kelvin wave. (**d**) Decadal variation of SST in the southeastern Arabian Sea for different seasons.

The current system associated with this second downwelling Kelvin wave transports the fresh, low-saline waters from BoB basin to the Southeastern AS (SEAS) during the post-monsoon season [58,67,98]. The intrusion of low-saline freshwater leads to the formation of temperature inversion causing significant increase in SST during November to February which in turn increase warming over SEAS [67,96,97,99,100]. The decadal mean SST

variation averaged over SEAS (Figure 7d) clearly shows the increase of warming trend during the post-monsoon season. A substantial increase in warming caused by strong influx of low-saline water into SEAS attributes due to increased strength of Kelvin wave propagation. Thus, this observed warming in the post-monsoon season play a major role in formation of Arabian Sea warm pool in the following season [68]. The westward propagating Rossby wave associated with the deepening of thermocline is radiated due to propagation of this Kelvin wave system [63,68]. The decadal variation in SSHA and the thermocline depth (D20) averaged over the latitudinal band (2° N–12° N) is shown in Figure 8. The clear westward propagation high SSHA is evident in each decade depicts the westward propagation of downwelling Rossby wave that is radiated due to the propagation of downwelling Kelvin wave. The decadal variation also shows that the increase in the strength of downwelling Rossby wave increases with each decade that could also play an important role in warming the southern regions of AS from December to April. In addition, the deepening of thermocline associated with the propagation of Rossby wave is also seen increasing in the recent decades. Increased strength in currents and heat transport from BoB to AS through the East India Coastal Current (EICC) result in warming during the winter season is discussed in the following section.

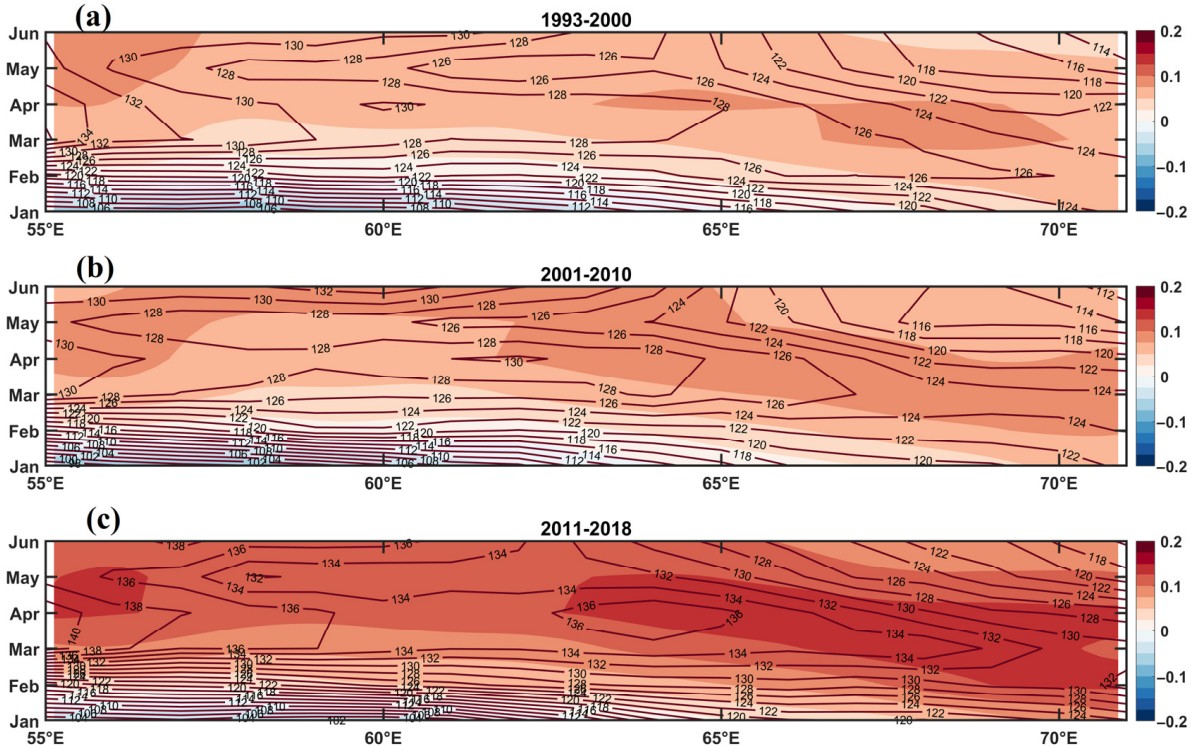

**Figure 8.** Decadal variation in SSHA representing the propagation of Rossby wave (shaded) along with the D20 contour averaged over the latitudinal band of 2° N–12° N.

The uncertain distribution of enhanced SST's and it inter basin tele-connections are still a question of debate today for the researchers [101]. The presence of an inter-basin teleconnection dipole between the warmer Atlantic and Indo-western Pacific with cooling over eastern Pacific is prominent during the past three decades [101,102].

During the past three decades, tropical sea surface temperature (SST) had shown dipole-like trends, with warming over the tropical Atlantic and Indo-western Pacific but cooling over the eastern Pacific. Variations in SST over the tropical Indian and Atlantic Oceans can modulate Pacific SST on interannual and decadal timescales. The Atlantic warming energize the easterly wind anomalies over the Indo-western Pacific as Kelvin waves and westerly anomalies over the eastern Pacific as Rossby waves. The wind changes induce an Indo-western Pacific warming exclusively focusing over the Arabian Sea through

the wind–evaporation–SST effect [103,104]. Arabian Sea warming is possibly one of the residual effects of this inter-basin teleconnections involving the propagation of Kelvin and Rossby waves. Study by [102] suggested the growth of stronger connection in the recent past between the tropical ocean basins based on the evidence from climate models. More enhanced warming trends over the Arabian sea can be expected in the near future.

### 3.3. Inter-Basin Scale Heat Transport

The EICC current system that flows equatorward from November in the BoB feeds the westward propagating Winter Monsoon Current (WMC) that flows south of Sri Lanka transporting low-saline waters from the BoB to the eastern AS. During the mature phase of WMC from December to February, the westward flowing WMC bifurcates into two branches, one flowing northward around the Lakshadweep high and merges into the poleward flowing West Indian Coastal Current (WICC). The other branch flows westward direction reaching the southern regions of the AS [58]. The schematic representation of circulation during winter monsoon is shown in Figure 1. The decadal variation pertaining to the current and heat transport from BoB is estimated for two regions (D1 and D2) as shown in Figure 1. One of the regions lies in the passage between the equator and Sri Lanka (D1, 2° N–7° N, 73° E–82° E) and the other lies in the AS mini warm-pool region (D2, 4° N–14° N, 68° E–78° E). The selection of the study region that is the passage between the equator and Sri Lanka is important, as the summer monsoon current and WMC flows through this region connecting the AS and the BoB basins. It is the only region that has a straight flow without having any meandering flow [58].

The decadal variation in currents and the total meridional, zonal heat transport averaged over $D_{700}$ for D1 and D2 are shown in Figures 9 and 10, respectively. Results indicate a strong decadal increase in current speed during the winter in both the domains. However, the current magnitude is seen strengthening in the recent years during winter as compared to the other seasons. Consistent with the current speed, the westward heat transport from BoB during winter also show a substantial increase during the recent decades. It is this decadal increase in heat transport which is one of the causative factors playing an important role in warming over AS during the winter season.

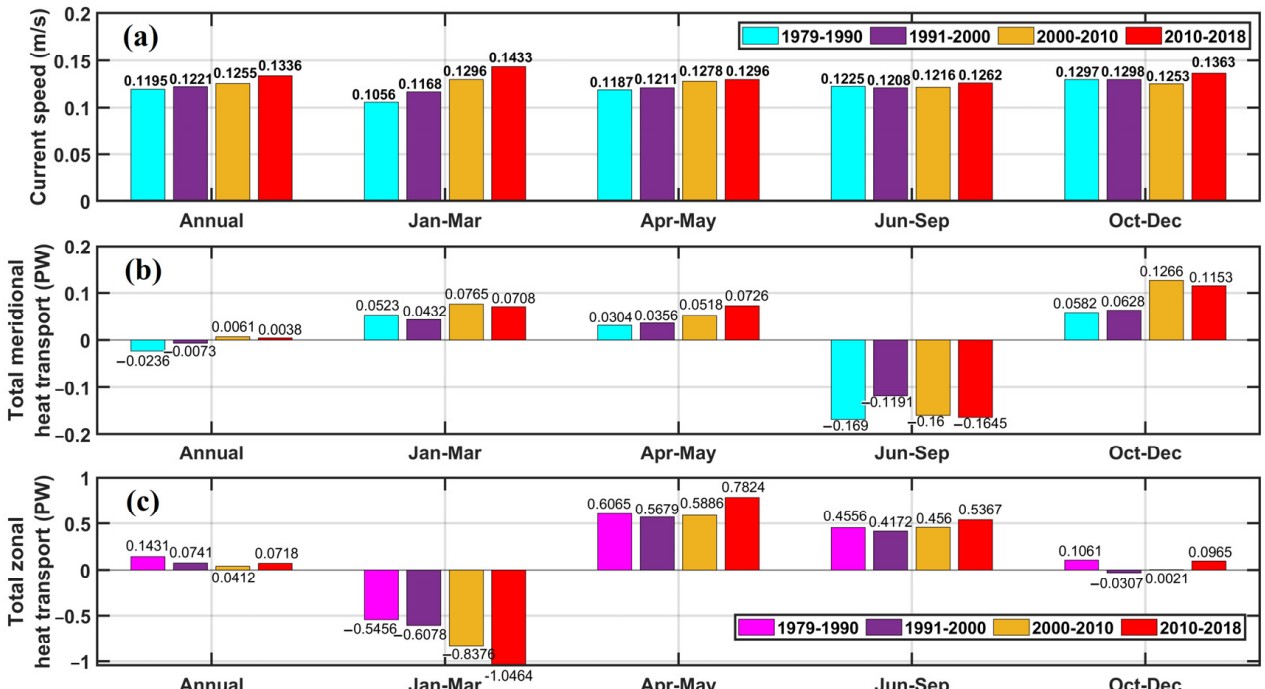

**Figure 9.** Decadal variation in (**a**) current speed, (**b**) meridional heat transport, and (**c**) zonal heat transport for different seasons in domain D1.

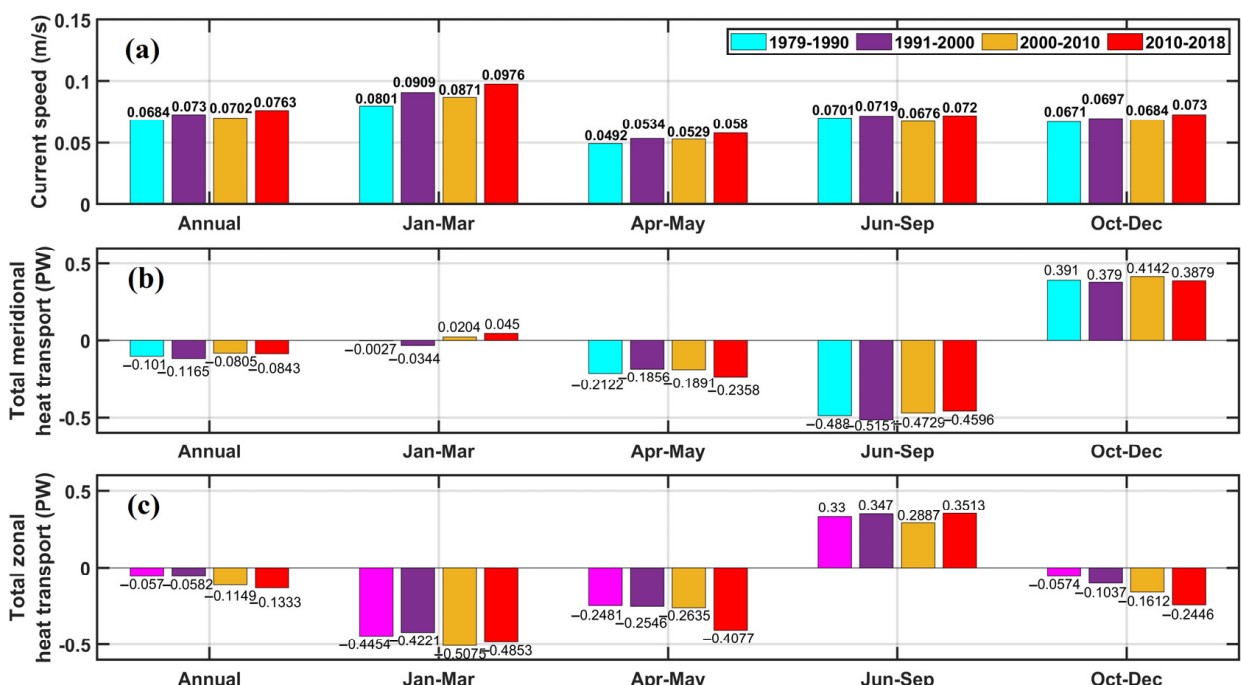

**Figure 10.** Decadal variation, the same as in Figure 9, but for domain D2.

### 3.4. Role of Surface Heat Flux and Entrainment in the SEAS

The physical processes responsible for increased warming trends in the SEAS are estimated using the heat budget analysis. The climatological monthly variation of surface heat flux, horizontal advection, vertical entrainment averaged over a period of 33 years are shown in Figure 11. Results clearly indicate that surface heat flux is the dominant factor that is responsible for warming in SEAS during the winter season, whereas vertical entrainment plays an important role during the monsoon season [105,106]. The trend analysis for each term in different seasons are shown in Figure 12. It is seen that for the pre-monsoon season, wherein the temperature tendency mostly depends on surface heat fluxes, a decreasing trend is noticed from 1985 to 2000. In the recent years, from 2000 onwards the surface heat flux exhibited an increasing trend, and that is consistent with increased warming over SEAS during this season. Vertical entrainment, the second dominant term showed a decreasing trend during the recent years indicating that the cooling effect caused by entrained water into the mixed layer also decreased during the recent years.

Surface heat fluxes play an important role in other seasons, with an increasing trend during the recent years. Vertical entrainment showed a decreasing trend indicating declined cooling in the mixed layer depths during both pre- and post-monsoon seasons, whereas during the winter (not significant) and monsoon periods an increasing trend is observed.

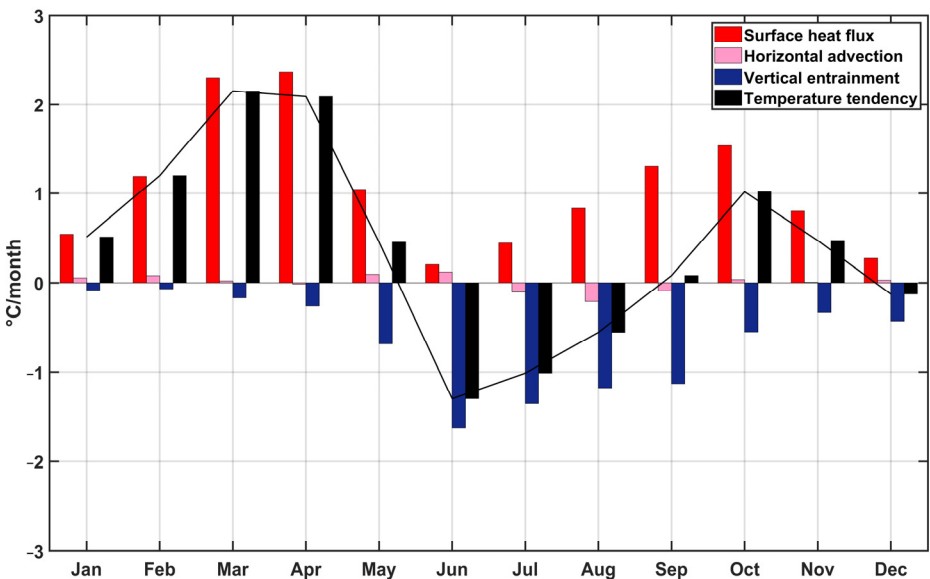

**Figure 11.** Monthly climatological variation of heat budget terms (°C/month) averaged in the SEAS.

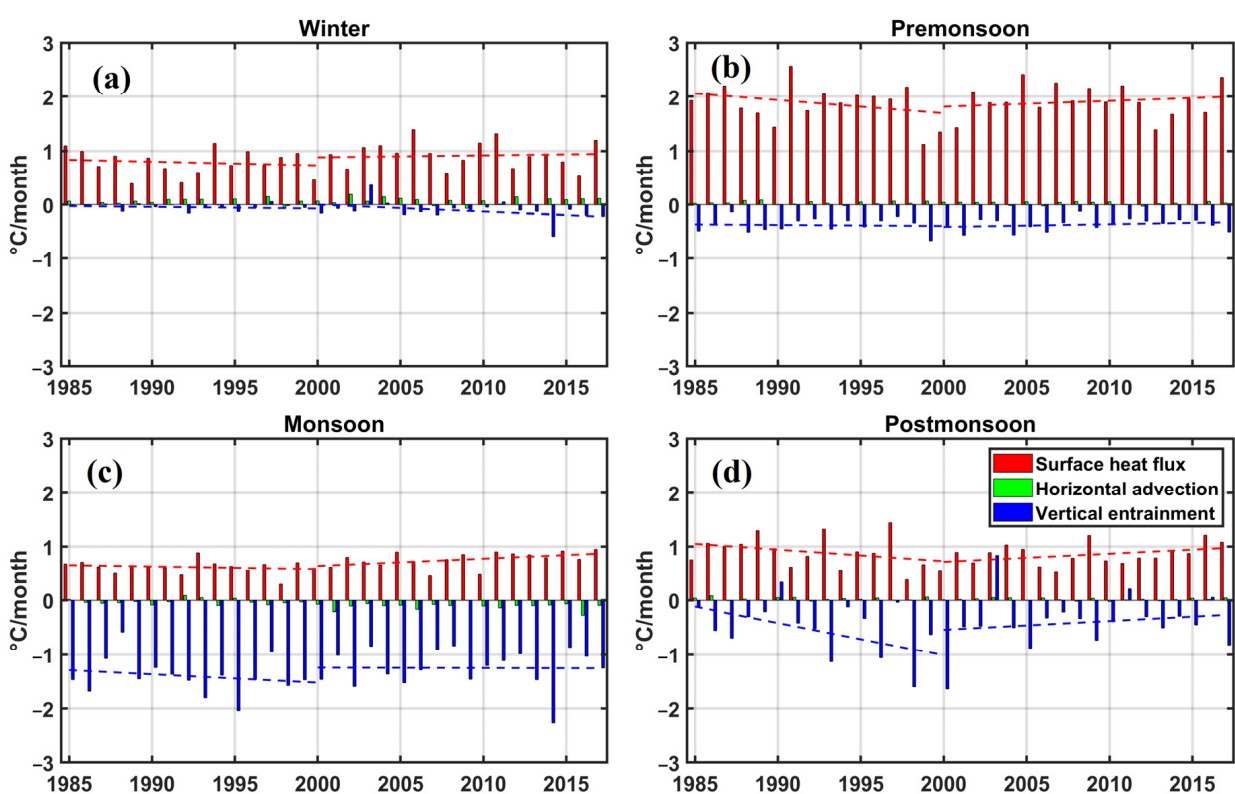

**Figure 12.** Trend analysis of heat budget terms in different seasons (**a**) winter, (**b**) pre-monsoon, (**c**) monsoon, and (**d**) post-monsoon.

*3.5. Frequency of Positive Indian Ocean Dipole (IOD) Months*

The positive (negative) IOD months are characterized as unusual warming (cooling) in the western IO and cooling (warming) over the southeastern IO regions. The positive IOD leads to drought conditions in Indonesia, lower rainfall than the average value over many parts of Australia and flood events in east Africa [107] and vice-versa for the negative IOD events. The frequent occurrence of positive IOD events might play an important role in increasing the weather extremes [108]. The total number of positive and negative months per year based on the Dipole Mode Index (DMI) is shown in Figure 13. It is evident that the

frequency of positive IOD months are increasing in the recent years, whereas the negative IOD months are declining. The results clearly suggest that the occurrence of positive IOD events are escalating in recent years could be associated with increased warming of AS in the recent years. The escalation of positive IOD events and warming of AS could be attributed to the recent increase in frequency of extreme weather events such as extreme severe cyclonic storms over the AS basin.

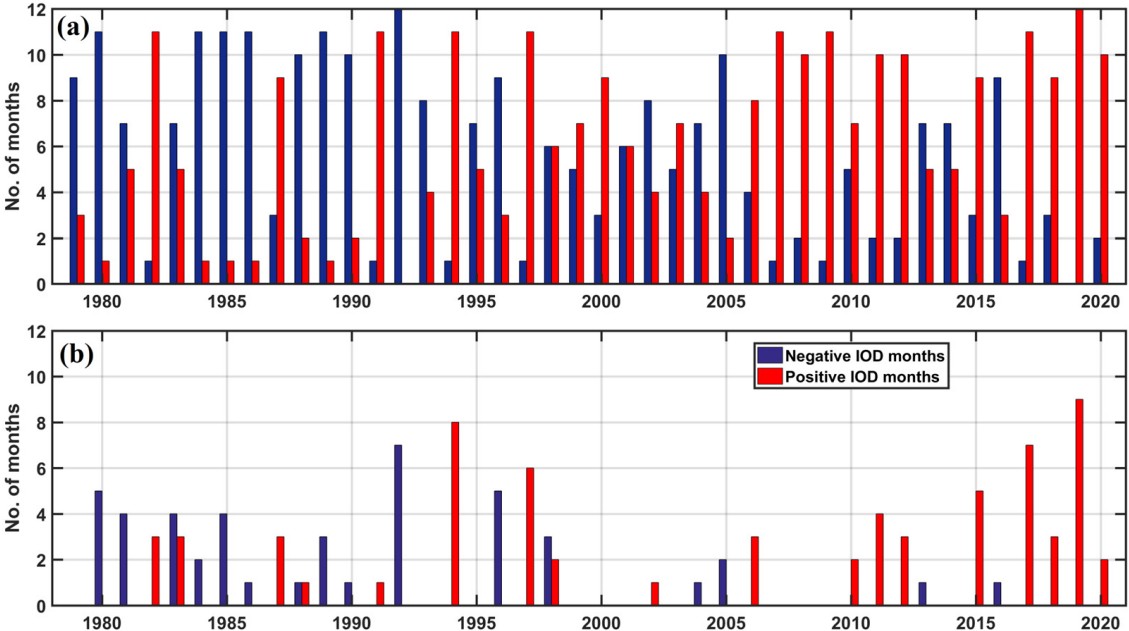

**Figure 13.** Total number of positive and negative months based on DMI. (**a**) The upper panel shows the total number of months having DMI < 0 (blue) and DMI > 0 (red). (**b**) Bottom panel; same as the upper panel, but for DMI > 0.4 and DMI < −0.4.

## 4. Conclusions

This study investigated the causative factors, possible physical mechanisms, and the characteristic feature of ocean parameters leading to the multi-decadal variations in the sub-surface ocean heat content over the North Indian Ocean region. It analyzed the historical observational data and thereby quantified the heat transport. Specifically, the study focused on the unique warming trends in the Arabian Sea region that recently experienced enhanced tropical cyclone activity. A detailed investigation was carried out using observational data products, derived heat transport, currents, temperature, salinity, outgoing longwave radiation, and the air-sea fluxes to quantify sub-surface warming trends in the past four decades. The causative factors and the underlying physical mechanisms responsible for increased warming have been reported. The main findings from this study are summarized as follows.

The study examined the causative factors and various physical mechanisms responsible for the warming trend in the Arabian Sea region using altimetric observations and reanalysis products. Among the different factors that contribute to warming in the Arabian Sea, the western regions of the Indian Ocean and Southern Indian Ocean region exhibited the highest variability in sub-surface ocean heat content. Warming in the deeper ocean layers is more significant compared to the upper layers in the Arabian Sea, wherein the trend is relatively imperceptible in the Bay of Bengal region. Sub-surface warming in the Arabian Sea is more prominent among the seven Indian ocean sub-domains considered in this study. Increased heat content is observed along with decreased trends in the Outward Longwave Radiation over the past 20 years. Whereas, a significant decreasing warming trend is noticed for the Bay of Bengal during this period. Significant evidence from the large-scale circulation features enhancing Arabian Sea warming are also observed during

the study. Altimetric observations reveal strengthening of second downwelling Kelvin wave propagation that causes warming over the eastern Arabian Sea attributed due to strong influx of low-saline waters from the Bay of Bengal leading to near-surface stratification. Rossby wave associated with the deepening of thermocline causes warming over the southern Arabian Sea during its propagation. Estimated heat transport through the conveying channel from Bay of Bengal to the Arabian Sea basin in the upper 700 m water depth increased rapidly in recent years during winter, and that plays a significant role in warming the southeastern Arabian Sea. Heat budget analysis reveals that the surface heat fluxes play a dominant role in warming the Arabian Sea during the pre-monsoon season. The increasing (decreasing) trend of surface heat fluxes (vertical entrainment) during 2000–2018 play a significant role in warming the southeast Arabian Sea. Additionally, increasing (decreasing) positive (negative) DMI months during the recent years has a relation with increased warming in the western Indian Ocean and southern Arabian Sea regions. The escalation of positive DMI months in recent years has a relationship with the increased occurrence of extreme weather events in the Arabian Sea.

The current study demonstrated the indication of warming and heat entrainment over the Arabian Sea basin especially over the southeastern subdomain region. The comprehensive ocean model analysis is warranted to quantify the incoming heat source to the Arabian Sea and outgoing flux rates, to measure the heat trapped inside ocean waters. The results can provide valuable insights to the future cyclonic activity over the basin.

**Supplementary Materials:** The following supporting information can be downloaded at: https://www.mdpi.com/article/10.3390/cli11020035/s1.

**Author Contributions:** All authors contributed to the study conception and design. Material preparation, data collection and analysis were performed by J.A. and V.S.G. The first draft of the manuscript was written by J.A., V.S.G., N.K.V., P.K.B. and all authors commented on previous versions of the manuscript. Conceptualization: P.K.B.; methodology: J.A., N.K.V., P.K.B. and M.K.D.; formal analysis and investigation: J.A., V.S.G., N.K.V. and P.K.B.; writing—original draft preparation: J.A. and V.S.G.; writing—review and editing: J.A., V.S.G., N.K.V., P.K.B. and M.K.D.; supervision: P.K.B. All authors have read and agreed to the published version of the manuscript.

**Funding:** The authors did not receive support from any organization for the submitted work. No funding was received to assist with the preparation of this manuscript.

**Institutional Review Board Statement:** Not applicable.

**Informed Consent Statement:** Not applicable.

**Data Availability Statement:** The datasets generated during and/or analyzed during the current study are available from the corresponding author on reasonable request.

**Conflicts of Interest:** The authors declare no conflict of interest.

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
