# Peer review of "Recent Warming Trends in the Arabian Sea: Causative Factors and Physical Mechanisms"

_climate, doi:10.3390/cli11020035_

Round 1
Reviewer 1 Report
This manuscript examines the factors that are responsible for the warming of the Arabian Sea. Possible facotrs that can raise OHC in the AS were discussed. But it is not clear that each factors contribute to what extent. In addition, I have some concerns about the results, as I pointed out below. Therefore, I recommend that the manuscript should be accepted after major revision.
Major commnents:
(1) Estimates of uncertainties of estimated linear trends in OHC, OLR, and so on should be shown in the manuscript. Although the authors discussed the estimated linear trends, readers cannot evaluate the results.
(2) The authors should show division of sub-domains on a map and discuss the validity. The authors show time series of OHC for each sub-dmain in Fig.1. But they should show that the division is reasonable for the following OHC analysis.
(3) Relation to OLR change is not convicing. The authors claimed the importance of the OLR change in the OHC change. However, I feel there is a gap between the OHC and OLR changes. Because OLR show temperature on the top of the atmosphere, which is closely related to atmospheric convective activity. Therefore, we cannot easily conclude that the OLR change is linked to the heat flux change at the sea surface. The authors should first examine net sea surface heat flux whether or not OLR variation is linked to heat flux change.
(4) The analysis of section 3.4 suggests that the results in section 3.3 are trivial. I cannot understand the importance of section 3.3.
(5) The frequency of positive (negative) IOD months are clearly increasing (decreasing). This is very interesting. The authors described the relation to cyclonic storms (L461-464). I also have expected that the frequency of the IOD is related to the warming of the subsurface water in the AS, the main target of this paper. If so, the authors should describe the relation to the warming.
Minor comments:
L27-28: "Increased OHC in AS show good correlation with decreased OLR in the past 20 years." As I commented above, there seems to be a gap. And we should be careful to examine correlation between linear trends.
L41-42: It is better to cite past papers. And this statement seems to be inconsistent with the following sentences of this paragraph. Please clarify.
L59: I cannot understand the expression "the less-active AS basin with enhanced convective activity." Please add explanations.
L55-93: This paragraph is too long. Please separate in to a few paragraphs.
L244: It is better to delete '' of rho and Cp.
Equation (2): two theta are multiplied. I guess either of them is v. Please correct this equation.
Table 2 and L283-284: For readability, it is better to show these sub-domains on an Indian Ocean map.
Figures 1 and 2: What is the value shown on the top of each panel? Regression slopes? For example, 4.64e+05 J/yr for Fig. 1a. Please explain in the captions.
L307-308 and other some lines: "the OHC distribution as a function of time." should be simply "the OHC time series."
Figure 6: SSH should be stated in the caption.
L397-401: I cannot follow locations of the current systems. Please their locations on a map such as Fig.8.
L397, 402: 'propagate(s)' should be changed to 'flows'
L433: I guess the authors discuss with Figure 12.
Reviewer 2 Report
This study examined recent warming trends in the Arabian sea, including causative factors and physical mechanisms. This recent warming trends may have important effects on tropical cyclone activities in this region. Some comments on the present manuscript are as follows.
1. Lines 483-519: There are many points listed in the main conclusions. It is suggested to combine and summarize the main conclusions. If there are too many small points listed, the conclusions are scattered and not concentrated.
2. Line 34-36: The Indian Ocean dipole is able to reproduce the observed increase in TC activity over AS. The statement of this sentence is not supported by relevant analysis in the following main text. It is inconsistent with the statement of the following main conclusions. It is suggested that the expression of this sentence should be changed to be consistent with the following main conclusions.
3. Line 115-160: This part could be simplified. There is too much content in the Introduction section.
4. Line 190: At the end of the introduction section, a short paragraph should be added to explain the purpose of this study and the overall structure of the manuscript.
5. Figure 1 and 2: The long-term change trend should be tested for significance to determine whether the trend is significant or not.
6. Figure 4: The trend does not seem significant in this figure.
7. Figure 5: What is the physical quantity represented by color shadings in this figure? More explanations should be added to this figure, otherwise it will be difficult for readers to understand.
8. Figure 6: What is the physical quantity represented by color shadings in a,b,c? Is there no letter mark (a,b,c,d) in the sub-figures?
9. Figure 7: What is the physical quantity represented by color shadings in this figure? More explanations should be added to this figure, otherwise it will be difficult for readers to understand.
10. Figure 9: Is there no letter mark (a,b,c) in the sub-figures?
11. Figure 12: Each sub-figure shall be marked with a sub-figure label (a,b,c,d).
12. Line 524: "could bring out can provide.." Here, "could bring out" should be removed?
13. Line 189: spreads -> spread.
14. Line 144: Ekman suction. Should be Ekman pumping?
15. The recent warming in the Indo-western Pacific may be attributed to the warming in the tropical Atlantic (Li et al., 2016). Some discussions on this point are suggested to be added.
Li X., Xie S P, Gille St, Yoo C, 2016: Atlantic-induced pan-tropical climate change over the past three decades. Nature Climate Changes, 6: 275-279.
Round 2
Reviewer 1 Report
Thank you for elaborate revision. I feel that the authors appropriately revised the manuscript.
Reviewer 2 Report
The authors have revised the manuscript according to previous comments.
Figure 5:There are four sub-figures in Figure 5, but only (a) and (b) are marked. All sub-figures need to be marked with letters (a,b,c,d). The meaning of the two sub-figures in the right column should be explained in the figure title.
